# Silver Nanowire Networks: Mechano-Electric Properties and Applications

**DOI:** 10.3390/ma12162526

**Published:** 2019-08-08

**Authors:** Hiesang Sohn, Chulhwan Park, Jong-Min Oh, Sang Wook Kang, Mi-Jeong Kim

**Affiliations:** 1Department of Chemical Engineering, Kwangwoon University, Seoul 01897, Korea; 2Department of Electronic Material Engineering, Kwangwoon University, Seoul 01897, Korea; 3Department of Chemistry and Energy Engineering, Sangmyung University, Seoul 03016, Korea; 4Material Research Center, Samsung Advanced Institute of Technology (SAIT), Samsung Electronics, Suwon 16678, Korea

**Keywords:** Ag nanowire networks, mechano-electric properties, flexible conductive electrodes

## Abstract

With increasing technological demand for portable electronic and photovoltaic devices, it has become critical to ensure the electrical and mechano-electric reliability of electrodes in such devices. However, the limited flexibility and high processing costs of traditional electrodes based on indium tin oxide undermine their application in flexible devices. Among various alternative materials for flexible electrodes, such as metallic/carbon nanowires or meshes, silver nanowire (Ag NW) networks are regarded as promising candidates owing to their excellent electrical, optical, and mechano-electric properties. In this context, there have been tremendous studies on the physico-chemical and mechano-electric properties of Ag NW networks. At the same time, it has been a crucial job to maximize the device performance (or their mechano-electric performance) by reconciliation of various properties. This review discusses the properties and device applications of Ag NW networks under dynamic motion by focusing on notable findings and cases in the recent literature. Initially, we introduce the fabrication (deposition process) of Ag NW network-based electrodes from solution-based coating processes (drop casting, spray coating, spin coating, etc.) to commercial processes (slot-die and roll-to-roll coating). We also discuss the electrical/optical properties of Ag NW networks, which are governed by percolation, and their electrical contacts. Second, the mechano-electric properties of Ag NW networks are reviewed by describing individual and combined properties of NW networks with dynamic motion under cyclic loading. The improved mechano-electric properties of Ag NW network-based flexible electrodes are also discussed by presenting various approaches, including post-treatment and hybridization. Third, various Ag NW-based flexible devices (electronic and optoelectronic devices) are introduced by discussing their operation principles, performance, and challenges. Finally, we offer remarks on the challenges facing the current studies and discuss the direction of research in this field, as well as forthcoming issues to be overcome to achieve integration into commercial devices.

## 1. Introduction

With the increasing demand for flexible optoelectronic and photovoltaic devices (e.g., wearable health-care devices and motion-recognition for AR (Augment Reality)/VR (Virtual Reality) devices), it has become critical to ensure the mechano-electric reliability of all components in such devices. However, traditional transparent conductive electrodes (TCEs) based on indium tin oxide (ITO) or metal thin films have limited flexibility and suffer from high processing costs, which undermine their application in flexible devices. The flexibility and stretchability of conventional ITO or metal thin-film electrodes can be achieved by modifying their structure into complicated (serpentine, wavy, or wrinkled) patterns or by the deposition of a very thin oxide layer on the flexible substrate [1,2,3,4,5,6,7,8,9,10]. Such modifications require lithography or patterning techniques, which introduce high costs and cause inconvenience [1,2,3,4,5,6,7,8,9,10]. 

In this context, there has been much interest in the application of nanostructured materials to flexible electronic and optoelectronic devices by convenient processes to overcome the shortcomings of conventional electrode materials (e.g., ITO and other oxides) [1,2,3,4,5,6,7,8,9,10]. As shown in Table 1, there are several proposed solutions of nanostructured materials, including carbon-based materials (carbon nanotubes (CNTs) and graphene), conductive polymer thin films, networks of metallic (gold, silver, copper, etc.) nanowires and patterned metal nanogrids [1,2,3,4,5,6,7,8,9,10]. Among them, silver nanowire (Ag NW) networks and Ag NW-related devices have been the focus of intensive research and development, owing to their good electrical, optical, chemical, thermal, and mechanical properties [1,2,3,4,5,6,7,8,9,10]. 

More specifically, as shown in Figure 1, Ag NW percolation networks are considered as promising alternative TCE materials because of their unique network geometry (one-dimensional (1D) structures), lack of dislocation activity, and high strength of metal NWs. Figure 1a displays the electro-optical properties of various electrode materials, where the electrical property was the contact resistivity, as measured by a four-point probe, and the optical property was the light transmittance, as measured by a haze meter. Despite the reciprocal relation between the electrical conductivity and optical transparency, Ag NW networks exhibit the closest values to those of a desirable transparent electrode. In addition, as shown in Figure 1b, Ag NW networks show the lowest electrical resistance under the smallest bending radius, suggesting excellent mechano-electric properties of Ag NW networks. Thus, such unique characteristics of Ag NWs render them an excellent choice for use in advanced flexible/wearable devices, which accommodate bending and stretching deformations [1,2,3,4,5,6,7,8,9,10]. Despite the superior electrical and optical properties of Ag NW networks to those of other conductive materials, the practical application of Ag NWs in electronic devices is limited owing to their poor mechano-electric stability under extended operation. Moreover, despite extensive studies on the electrical and mechanical performance of Ag NW networks, there has been no comprehensive review of the mechano-electric properties and the applications of Ag NW networks.

This review presents the recent developments in the mechano-electric properties of Ag NW networks toward various flexible devices by summarizing the strategies and notable cases in the recent literature. In addition, the challenges (e.g., Ag NW reflectivity, NW junction instability, chemical instability, nonuniform electron density, and pattern properties) in the current studies, the research direction, and future issues in the field are discussed. More specifically, as the key properties required for flexible conductive electrode materials are mechano-electric and electrical/optical robustness under repeated deformation, we focus on the mechano-electric property of Ag NW networks from the viewpoint of their application in various practical electronic devices. 

In Section 2, we discuss Ag NW networks by presenting (1) the synthesis of NWs in solution processes, (2) the arrangement of Ag NWs into percolation networks through various coating processes, and (3) their characteristics including chemical, electrical, and optical properties. Specifically, we not only present the synthesis of Ag NWs through various synthetic methods, but cover deposition processes used to form Ag NW networks from lab-scale coating processes (drop casting, spray and spin coating, etc.) to commercialized processes (slot-die and roll-to-roll (R2R) coating). The individual and network characteristics of Ag NWs also discussed by focusing on their electrical and optical properties, which are governed by the percolation and electrical contacts of the Ag NW networks. 

In the Section 3, we review the mechano-electric properties of individual and integrated Ag NW networks by demonstrating reliable electronic systems with mechanical flexibility. Note that while mechanical characteristics such as the flexibility of Ag NW networks are primarily governed by the mechanical robustness of individual NWs, the mechano-electric properties of NW networks are affected by the static mechanical properties, as well as the electrical percolation and connections of the networks. Next, we discuss the extended mechanical stability (bending fatigue stability) of Ag NW networks under long-term cycles. As the bending fatigue resistance is the commonly encountered failure mode, the stability, integrity, and lifetime of NW-based devices can be affected by the mechanical fatigue behavior of the electrode. The improved mechano-electric properties of Ag NW networks are discussed by presenting the efforts made to enhance the performance of Ag NW network-based flexible electrodes through post-treatment and secondary structural hybridization [1,2,3,4,5,6,7,8,9,10]. 

In the Section 4, we provide an overview of various Ag NW network-based flexible devices under dynamic cyclic loading, as Ag NWs are expected to be suitable for versatile wearable applications based on their superior mechanical, electrical, and optical properties. Specifically, we cover various Ag NW network-based flexible devices (optoelectronic and electronic devices) requiring stable fatigue resistance to failure after extended cyclic deformation. Although the demonstrated devices are prototypes, it is still meaningful to introduce such devices before integration into commercial products, because of their high potential toward real applications. 

We believe that this review provides a design guide to construct flexible electronic devices based on Ag NW networks and their composites with high mechano-electric reliability. 

## 2. Silver Nanowire (Ag NW) Networks 

In this section, we discuss the fabrication methods and the characteristics of Ag NW networks. It should be noted that successfully fabricating flexible transparent conductors requires materials with superior optical transparency, electrical conductivity, and mechanical flexibility. In these respects, Ag NW networks have been prepared that exhibit suitable properties as electrode materials for the flexible devices [5,6,7,8,9,10,11,12,13,14,15,16,17,18,19,20,21,22,23]. 

### 2.1. Formation of Ag NW Networks 

The formation of a network structure of Ag NWs on flexible plastic substrates is essential for the successful integration of Ag NWs into various applications of flexible TCEs, as an optimized percolation network of Ag NWs is crucial to achieving the target mechano-electric properties of flexible TCEs [1,2,3,4,5,6,7]. It should be noted that Ag NW networks can demonstrate higher electrical conductivity at large strains than can a single NW, without deformation induced electrical conductivity decrement [1,2,3,4,5,6,7,8,9,10]. Such Ag NW networks enable the achievement of desired electrical and optical properties of TCEs constructed by coating the Ag NWs on flexible substrates.

Considering the dependence of the conductive electrode performance upon the coating technique, it is required to develop simple, reliable, and cost-efficient fabrication techniques (deposition processes) to form Ag NW-based conductors with high flexibility and conductivity [1,2,3,4,5,6,7,9,10]. Specifically, a large-area Ag NW network enables highly transparent (~90%) and conductive conductors, where the properties are dependent on the deposition technique and wire geometry [1,2,3,4,5,6,7,9,10]. The networks of Ag NWs should be readily and cheaply prepared, as well as scalable to large are through solution processing techniques. Nevertheless, it is often difficult to achieve thin films prepared through colloidal solutions or precursor deposition with highly reproducible opto-electrical and mechanical properties. 

As typical solution coating processes are to be compatible with low-temperature processes (<200 °C), they do not require expensive equipment [1,2,3,4,5,6,7,9,10]. Hence, there have been many studies on the production of Ag NW electrodes through simple, reliable, and cost-efficient deposition techniques. 

As summarized in Scheme 1, several fabrication methods are available for Ag NW network-based TCEs for flexible devices. Most fabrication techniques are compatible with solution deposition processes such as spray coating, drop casting, spin coating, rod-coating, dip coating, vacuum filtration, slot-die coating, and R2R coating, as solution-processed electrodes have already demonstrated their ability to be integrated into optoelectronic devices [8,9,10,11,12,13,14,15,16,17,18,19,20,21,22,23]. 

#### 2.1.1. Drop Casting 

Drop casting is the simplest method to produce an Ag NW network, which consists of dropping a NW-dispersed solution on a target substrate followed by drying the residual solvent. For the solution precursor preparation, water or alcohols (ethanol, methanol, or isopropanol) are employed as the dispersion solvent [11]. The drop-casting procedure, as illustrated in Figure 2A, can be applied to prepare Ag NW network-based electrodes with unique physical properties using regulated precursor solutions. For instance, the optical transmittance of the drop-casted Ag NW network largely depends on the Ag NW solution’s concentration or its evaporation rate after coating [11]. In addition, the Ag NW network (Figure 2B(a) formed by drop casting exhibits similar surface roughness for both glass (Figure 2B(b) and polyimide (PI) (Figure 2B(b) substrates. 

Despite the facile fabrication process of drop casting, the electrodes prepared by this method exhibit limitations for practical electro-optical device applications. For instance, the active process area/range is highly restricted. In addition, the drop-casted film thickness is uncontrollable because of spatial inhomogeneities (or uneven distribution of Ag NWs) of drop-casted NW networks caused by the surface tension of the solution and the self-aggregation of NWs by the “coffee ring effect” on the substrates [11]. What is more, as shown in Figure 2B(b,c), re-deposition of Ag NW by drop casting further aggravates their surface morphology (or makes uneven vertical distribution of NWs), leading to non-uniform electrical and optical property of Ag NWs network [11].

In order to address the current challenges, several modified drop-casting processes have been proposed. For instance, the hybrid structure of a drop-casted Ag NW network in a UV-curable polymer (poly-acrylate resin) can be prepared by the sequential deposition, curing, and peeling-off of the UV-curable polymer [11]. The properties of the composite will be discussed in Section 3. 

#### 2.1.2. Spin Coating 

Spin coating is a reproducible and scalable process to form Ag NW network films with uniform thickness and radial anisotropy. Typically, as shown in Figure 3A, spin-coated Ag NWs film can be formed by consecutive processes of applying coating solutions onto a substrate, followed by spin coating at a certain rotating speed (rpm) and time to achieve the desired thickness or film quality. After spin coating, the Ag NW network is formed on the target substrate after the removal of most of the coating solution [12,13]. The properties of Ag NW network films can be regulated by judicious control of the coating solution (viscosity, volatility, and concentration), substrate, and rotational speed. As shown in the schematic illustration (Figure 3B(a), a composite film of Ag NWs (ZnO/ITO/AgNW/ITO) also can be constructed through successive spin coating processes. Figure 3B(b) displays FE-SEM images of composite electrodes, showing good dispersion of Ag NWs sandwiched between oxides (ZnO and ITO) [12,13].

#### 2.1.3. Bar Coating 

Bar coating is the most widely used method to prepare Ag NW networks, in which a precursor solution is directly applied to a target substrate with a Mayer-rod bar. As shown in the process schematic (Figure 4a), Ag NW networks are formed by sweeping and spreading out an Ag NW solution with a Mayer-rod across the substrate [14,15,16]. The thickness of the Ag NW networks is determined by the initial wet-thickness applied with the Mayer-rod and the concentration of the Ag NW solution layer on the substrate. A relatively fast solvent evaporation rate (compared with drop casting) can effectively prevent the uneven thickness and local agglomeration of Ag NWs caused by the coffee ring effect [14]. As demonstrated by Lee et al., scalable and flexible TCEs can be produced by optimizing coating parameters (slit size of the rod, solvent), precursor solution (concentrations, additives), and other aspects [14]. As shown in Figure 4b, Ag-NW TCEs without TiO_2_ deposition were prepared with different coating cycles. The density of NWs on the film increases in proportion to the coating cycle. As shown in Figure 4c, the sheet resistance decreases with additional bar coating cycles (1000, 250, 100, and 20 Ω/sq. for the 3rd, 4th, 5th, and 6th cycles, respectively).

Furthermore, the Ag NW network produced by bar coating can be post-treated with a peel-off process to obtain a free-standing film. As demonstrated by Ji et al., Ag NW network-based flexible TCEs were deposited by bar coating, followed by subsequent curing and peel-off processes, which led to the formation of a partially embedded hybrid structure of Ag NW on the target substrate [15]. In addition, as the bar coating yields alignment of the Ag NW networks along the dragging direction of the rod, the aligned Ag NW network structure can show improved conductivity and surface uniformity through NW disentanglement and the reduction of NW–NW junction resistance [15]. 

#### 2.1.4. Vacuum Filtration and Transfer (VFT) 

Since the pioneering work by De et al., the vacuum filtration and transfer (VFT) process has enabled the fabrication of free-standing Ag NW-based flexible TCEs [17]. As the losses of Ag NWs to form NW networks are much smaller than those of other processes, they are actively used to prepare flexible TCEs. As shown in Figure 5A(a), Ag NW networks can be prepared by VFT by the following steps: (1) filtration of Ag NW solution onto a filter, (2) formation of a NW network layer on the filter, and (3) transferring the network layer to the target substrate (polydimethylsiloxane (PDMS), PET, or paper). Afterwards, as shown in a photograph and SEM images of Ag NW felt after annealing and PDMS infiltration (Figure 5A(b)), it was found that a thick layer of free-standing Ag NW networks was formed without deterioration after the thermal and pressured processes. It should be noted that the film quality (areal density) of the Ag NW network can be easily controlled by judicious adjustment of the concentration and amount of the NW precursor solution. The final product (Ag NW network-based TCE) can be obtained by transferring the as-formed NW network layer onto the target substrate. In a similar method to VFT (Figure 5B(a)), Ko et al. demonstrated Ag NW network-based TCEs on PDMS by a successive multistep growth (SMG) process, which exhibited good optical properties (Figure 5B(b)), flexibility, and excellent stretchability [18]. The excellent adhesion and mechanical stability of the Ag NW TCE can be attributed to the unique composite structure of Ag NWs networks partially embedded in the PDMS (Figure 5B(c)). However, despite many advantages of VFT techniques to prepare a uniform thin film of Ag NWs, the process is not compatible with large-area films, as the transferring areas are limited by the membrane filter size.

#### 2.1.5. Spray Coating

Spray coating allows uniform, large-area deposition of Ag NWs on various substrates, owing to its simple and scalable processes of depositing multiple layers of material [19,20,21]. The films obtained by spraying coating are usually more homogeneous and tend to form much more uniform networks than those produced by other methods. As shown in Figure 6a, Ag NW networks are formed by spray coating an Ag NW solution of onto a target substrate (PET). As shown in Figure 6b–e, the as-formed Ag NW network exhibits good electrical (sheet resistance: 11.3 Ω/sq.) and optical properties (transmittance: 81.7%) (Figure 6b,d), as well as uniformly distributed NWs (Figure 6c,e). The simplicity of the spray-coating process enables the formation of large-area NW network layers and it may be combined with mass-production processes such as R2R [19,20,21]. For instance, a combined spray coating of Ag NW and other layers (e.g., PEDOT:PSS) enables the achievement of hybrid-structure high-performance TCEs [19]. Akter et al. reported a high-performance flexible conductor prepared by spray coating, in which the adhesion force between the Ag NW and PDMS was effectively enhanced by the surface treatment with polydopamine on the PDMS substrate [20]. Madaria et al. further demonstrated soft-patterned Ag NW electrodes by spray-coating-assisted transfer using PDMS as a stamp. Specifically, the spray-coated Ag NW layer on a PDMS substrate was transferred to a PET substrate through simple press contact, owing to the weak adhesion force between the PDMS and Ag NWs [21].

#### 2.1.6. Slot-Die Coating 

Scalable and large-area deposition processes are required for the commercial production of Ag NW networks. In this context, slot-die coating is an appropriate approach to deposit a thin film of Ag NW on the target substrate, and it can be easily integrated into scalable processes such as R2R coating. Slot-die coating offers excellent coating uniformity across the coating surface, and it can deposit thin films with various thickness ranges (from nanometers to micrometers). Moreover, this method can deposit a wide range of coating solutions with various concentrations and viscosities by controlling the deposit rates in the range of ~cm/s to ~m/s [22].

As shown in Figure 7, Ag NW network films can be prepared using a continuous R2R slot-die coater under atmospheric conditions. The flow of Ag NW precursor solution at the slot die lip is fed to the target substrate to control the density of Ag NWs on the substrate (Figure 7a). As a result, the continuous R2R-compatible slot-die coating process combined with over-coating layer deposition allows the production of high-performance TCEs based on Ag NW network films (Figure 7b) [23]. Owing to the excellent processability offered by slot-die coating, this method has been employed as a transitioning process from lab-scale to scaled-up production of thin films [23]. In particular, this method is employed to determine the feasibility of transferring NW solutions for scalable device fabrication. 

Slot-die coating is further scalable for large-area Ag NW film production through R2R-based continuous processing by using the mechanical flexibility of Ag NW films (Figure 7b) [22,23]. During coating, the film substrate initially unwound from a roll passes through the coating machine, followed by rewinding onto another roll after film coating. The quality of film produced by R2R processes can be affected by many factors such as tension, speed, substrate cleanness, and static electricity. To improve the commercial compatibility, there have been many studies on the development of novel R2R systems to scale up the Ag NW films produced in the lab. For instance, as reported by Hösel et al., the performance of flexible electronic devices is highly dependent on the R2R coating systems, because the systems are difficult to unify into universal process [22,23]. 

#### 2.1.7. Printing 

Conventional patterning techniques (direct laser ablation, shadow mask, chemical etching) based on the photolithographic processes are not compatible with the preparation of Ag NW-based flexible TCEs [24,25,26,27]. As an alternative, various printing approaches have been introduced, such as spray coating and drop-on-demand systems. However, as these processes are not appropriate for commercial production, a scalable method to fabricate Ag NW network films is needed. 

In this context, direct printing techniques allow the production of flexible conductive electrodes without masks and templates (Figure 8) [24,25,26,27]. In particular, as illustrated in Figure 8a, electro-hydrodynamic jet (EHD) printing provides an improved resolution and finer patterning compared to that of conventional printing [24,25]. As reported by Park et al., the electrical conductivity of the TCE could be improved while retaining the optical transmittance of the pre-patterned Ag grids. Moreover, the flexible TCE can be formed by Ag grids with various line widths and spacings on the target substrates by EHD printing [24]. For instance, as shown Figure 8b,c, Cui et al. reported large-scale Ag NW pattern printing on PDMS by EHD printing [25]. However, despite the direct and facile patterning by the printing technique, the process requires several complicated procedures to obtain the desired patterning through judicious control of the nozzle diameter, working distance, applied voltage, and stage movement [24,25]. 

Apart from EHD printing, there are other printing techniques by which it has been attempted to produce high-quality Ag NW networks. These include PDMS stamp-assisted dry transfer printing, blown-bubble film coating, microfluidic assembly, electronic/magnetic field-assisted assembly, convective force assembly, layer-by-layer self-assembly, gravure coating, and electrospinning [24,25,26,27,28]. 

#### 2.1.8. Post-Processing

In addition to deposition processes of Ag NW, post-treatments (compression, thermal treatment, UV curing, chemical welding, and drying) have a great impact on the performance of Ag NW films [29,30,31,32]. Specifically, such post-treatments can be carried out on produced films to further improve the physical properties or functions of the Ag NW films by controlling the ordering and arrangement of the NWs. These additional treatments are conducive to improving the device performance of Ag NW by reducing the surface roughness and NW contact resistances. For instance, Lai et al. reported Ag NW-based TCEs with moth-eye nanostructures, whose optical transmittance was greatly enhanced by combining the R2R process with a post-treatment [29]. Bai et al. presented a lamination process by embedding NWs within a polymer or by sandwiching NW networks between ZnO layers [24]. Recently, Chung et al. and Kim et al. demonstrated that embedding NWs into a polymer yields reduced junction resistances and smoothed surface morphology, leading to improved mechanical substrate adhesion and retained mechanical flexibility [31,32]. The junction resistances of Ag NW networks can be further reduced by annealing with spatially selective plasmonic welding or thermal treatment under vacuum [33,34,35]. For the wide application of flexible devices, the deposition techniques for Ag NW networks introduced in this section should be further commercialized to industrial production levels. 

### 2.2. Characteristics of Ag NW Networks

Ag NW-based flexible TCEs require several parameters (chemical composition, morphology, density of NW network) for the efficient incorporation of NWs into various flexible devices. In this section, we discuss certain important fundamental properties (other than mechano-electric properties) of Ag NWs, such as the electro-optical properties and electrical stability. 

#### 2.2.1. Electro-Optical Properties

Research on Ag NWs has been focused on the enhancing the optical and electrical properties of Ag NW network-based TCEs by optimizing the NW networks [1,2,3,4,5,6,7,8,36,37]. As the Ag NWs are connected in various ways to form a network, the electrical and optical properties of network film can be estimated based on the assumption of open or closed paths in the object film [1,2,3,4,5,6,7,8,36,37]. A percolating system of an Ag NW network can determine the density of open pathways where a phase transition occurs. The association of connected pathways from one side to the other results in the phase transitions in percolating media of NW networks. 

As the optical and electrical properties mainly determine the performance of optoelectronic devices, the optical transparency and conductivity of Ag NW networks are two major characteristic parameters for NW-based TCEs. Typically, the transparency of the film can be quantitatively expressed as a function of the transmittance (***T***) and the sheet resistance (***R_s_***) of the film ($T=e^{-\alpha /\sigma_{DC,B}R_{s}}$, ***α***: absorption coefficient, ***σ_DC_***_,***B***_: bulk DC conductivity of the film) [36,37,38,39]. 

However, the optical transmittance of metallic NW-based TCEs can be defined in different way, where the free space among NWs plays an important role determining the transparency of Ag NW-based TCEs [36,37,38,39]. As the free space is related to the density of the NWs in the film, the transparency of the TCEs can be described in terms of the free space. As represented by the relation between the transmittance and ***R_s_*** in Equation (1), the transmittance of TCEs depends on the density of the Ag NWs [36,37,38,39]. 

In the bulk-like regime, the transmittance of Ag NWs can be expressed as follows:
(1)$$T={\left(1+\frac{Z_{0}}{2}\sigma_{O_{P}}t\right)}^{-2}$$
(***σ_Op_***: optical conductivity (***σ_Op_*** = ***α/Z*_0_**), ***Z*_0_**: impedance of free space) 

It is essential to determine the critical density ***n*_c_** in the Ag NW network percolating system. ***n*_c_** is defined as the density at which the probability of finding over 50% in a given percolating system, where ***n_c_*** is heavily dependent on the system geometry, lattice type, and object type [36,37,38,39]. The DC conductivity of a NW network film is nonlinearly related to the difference between the density of the NWs per unit area (***n***) and the percolation threshold (***n*_c_**), as shown in Equation (2):(2)$$\sigma_{DC}\propto {\left(n-n_{c}\right)}^{m}\left(m:\;4/3\right)$$

In addition, the percolation threshold (***n*_c_**) can be determined by the length of the NWs (***L_NW_***) using the equation $n_{c}=5.63726/L_{NW}{}^{2}$ [36,37,38,39,40,41,42,43].

A percolation network of Ag NWs is illustrated in Figure 9 [40]. The resistance of the Ag NW network decreases with time (Figure 9a) where the percolation network is activated by thermal annealing (Figure 9b). This result indicates that the formation of a percolation network of Ag NW reduces the resistance [40].

As demonstrated, adding some volume fraction (*x*) of conductive materials to poor conductors increases the conductivity for the volume fraction (*x*) range above a critical value (*x*_c_) [36,37,38,39,40,41,42,43]. In this regard, the effect of NW geometry on the electro-optical properties has been examined to further optimize the device performance of Ag NW-based TCEs. As examined by Pike and Seager, the network of longer NWs is more conducive to enhancing the conductivity of TCE films [36,37,38,39,40,41,42,43]. Similarly, as confirmed by Sorel et al., Ag NW networks exhibit linearly proportional conductivity to the NW length, whereas the optical conductivity is almost independent of the NW length [36,37,38,39,40,41,42,43]:
(3)$$T={\left(1+\frac{Z_{0}}{2R_{s}}\frac{\sigma_{O_{P}}}{\sigma_{DC}}\right)}^{-2}$$

The experimental data from different materials with transmittance values from 0 to the critical values, depending on the structure of the materials, can be fit. Beyond the critical values, the data deviate from the fitting curves, and percolation theory should be applied.

The DC conductivity of a NW network film can also be expressed using the film thickness, as shown in Equation (4) [36,37,38,39,40,41,42,43]:
(4)$$\sigma_{DC}\propto {\left(t-t_{c}\right)}^{n}\left(t_{c}:\;\mathrm{threshold}\;\mathrm{thickness},\;n:\;\mathrm{percolation}\;\mathrm{exponent}\right)$$

To fabricate high-conductivity NW networks for industrial applications, *t* must be greater than *t*_c_ [41,42,43]. Based on this, Coleman and co-workers further defined the relation between *σ_DC_* and *σ_DC_*_,*B*_ using Equation (5) [41,42,43]:
(5)$$\sigma_{DC}=\sigma_{DC,B}{\left(\frac{t}{t_{min}}\right)}^{n}\;\left(t_{\min}:\;\mathrm{thickness}\;\mathrm{of}\;\mathrm{Ag}\;\mathrm{NWs}\;\mathrm{network}\;\mathrm{film}\right)$$

Coleman et al. further used *n* to define a percolation figure of merit (∏), which indicates the relation between the transmittance (*T*) and the sheet resistance (*R_s_**)* of a TCE, as shown in Equation (6) [41,42,43]:
(6)$$\prod =2{\left[\frac{\frac{\sigma_{DC.B}}{\sigma_{O_{P}}}}{{\left(Z_{0}t_{min}\sigma_{O_{p}}\right)}^{n}}\right]}^{\frac{1}{n+1}}\;\left(T={\left[1+\frac{1}{\prod }{\left(\frac{Z_{0}}{R_{s}}\right)}^{\frac{1}{n+1}}\right]}^{-2}\right)$$

Figure 10a shows the optical transmittance (at 550 nm) of an Ag NW network as a function of its sheet resistance; using the experimental data with Equation (6), it is easy to obtain the values of ∏ and ***n*** [36,37,41,42]. ∏ is a dimensionless number that reflects the values of the sheet resistance and the transmittance, and ***n*** reflects the junction resistance in the NW network. It is found that the size effects of Ag NWs are closely associated with percolation in transparent conductors. The curve of optical transmittance (***T***) as a function of sheet resistance (***R_s_***) in Figure 10b shows that TCEs do not exhibit discernible differences with respect to the deposition method. This means that the electro-optical performance does not depend on the coating process, but rather on the geometry of the constituent materials. Different film deposition techniques are compared, which indicates that the properties of the network are independent of the fabrication method. The green line represents a fit to the bulk regime, and the orange line shows the fit for the percolative regime [36,37,41,42].

#### 2.2.2. Electrical Stability 

The electrical stability of an Ag NW network is important as the conductivity of Ag NW-based TCEs decreases because of electrical stress arising from electromigration or elevated temperature caused by Joule heating [1,2,3,4,5,6,7,8,43,44,45]. Specifically, the increased contact resistance of Ag NWs by high current density at the junction results in Ag electromigration and Joule heating of the network [44,45]. Under such conditions, vacancies in the Ag NWs can further increase the vacancy concentration and stress gradient of NW network by the rapid sweeping between the electrodes, leading to an avalanche-breakdown of the circuit beyond certain value of the local stress gradient [44,45,46,47]. Additionally, under similar conditions of high current density at the NW junction, the increase in the local temperature of the Ag NWs at the network junction to more than 300 °C leads to the melting (or evaporation) of Ag NWs due to Plateau–Rayleigh instability [35,44,45].

Such an electrical instability of NWs can be improved by junction resistance regulation, which reduces the local current density and the Joule effect [44,45,46]. Specifically, the electrical stability of Ag NW-based TCEs can be improved by effectively retarding the atomic surface diffusion of Ag atoms before the junction deterioration occurs through chemical treatments, electrical welding, laser sintering, or the deposition of extra layers, which enhances the mechano-electric reliability of the NWs [1,2,3,4,5,6,7,8,43,44,45,46]. 

## 3. Mechano-Electric Properties of Ag NW Network

The mechano-electric properties of Ag NW-based TCEs are affected by their network geometry and high strength of the NWs [48,49,50]. Specifically, Ag NWs exhibit superior mechano-electric reliability in comparison with metal thin films. In this context, it has been increasingly important to characterize and understand the mechano-electric properties of Ag NWs and their networks. This section focuses on the mechano-electric properties of Ag NW and their networks to provide the necessary information to meet the needs of flexible optoelectronic device applications. 

### 3.1. Mechanical Property of Nanowires 

Since the pioneering work by Galt and Herring regarding Sn whiskers (diameter: ~20 μm) that showed 10 times higher elastic strain (2–3%) than that of their bulk counterparts, the research on the fabrication and mechanical characterization of nanostructured materials (<1 μm) has been intensively pursued through various nanomechanical tests (uniaxial deformation and bending tests) [49,50,51,52]. In addition, there have been developments for the mechanical characterization of NWs through atomic force microscopy (AFM) and electron microscopy (EM), focused ion beam SEM technology (FIB-SEM), and the integration of micro-electro-mechanical systems (MEMS) into electron microscopes (EMs) [48,53,54]. Recently, an in situ uniaxial micromechanical device has been developed and utilized for the mechano-electric characterization of 1D Ag NWs, and bi-axial micromechanical devices are developed for the 2D nanomaterials [48,55]. Through the mechanical characterizations, two representative factors affecting the mechanical properties of NWs have been suggested, which are the microstructural size and crystalline structures. 

#### 3.1.1. Microstructural Size 

The performance of NW electrodes is closely related to their dimensions. That is, NWs exhibit the size dependence in the mechanical strength aspects of the Young’s modulus, fracture strength, and the yield strength [56,57,58]. Likewise, the strength values can scale with dimensions, such as the sample size or microstructural length, a phenomenon known as the mechanical size effect [56,57,58]. Specifically, the mechanical strength of a nanomaterial approaches its theoretical value, because of the relatively low intrinsic defect density in small-scale samples, which leads to very high strengths [56,57,58,59,60]. In contrast, when the size of the material is large, the surface energy barrier to be overcome also increases, and the generation of twins is suppressed. For instance, the yield strength of Au nanostructures (NWs and whiskers) under uniaxial stress exhibits a close-to-theoretical yield strength (~1 GPa) that is much higher than that of the bulk polycrystalline metal (~10 MPa) [59,60]. Recent uniaxial tensile tests also indicate a transition in the mechanical properties with decreasing sample size from ductility with little work hardening to brittleness [56,57,58,59,60]. In addition, under decreased material dimensions, the role of free surfaces becomes increasingly important in their deformation and fracture, owing to the increased surface-to-volume ratio. Specifically, as confirmed by tensile tests of Au NWs, the increased surface energy suppresses deformation twinning of NWs [59,60]. 

#### 3.1.2. Crystalline Structure 

The crystalline structure also affects to the mechanical properties of Ag NWs. Specifically, single-crystal (SC) and twinned NWs show different tensile behaviors, whereas the elastic modulus of twinned NWs is similar to that of SC NWs [54,61,62]. In addition, the dislocation nucleation of NWs takes place at the intersection of twin boundaries with the free surface, exhibiting site-specific behavior at the yield point [54,61,62]. All partial dislocations of NWs are hindered by twin boundaries under the maximum flow stress. Such partial dislocations cannot escape the crystal, in stark contrast to the behavior of SC NWs [54,61,62].

There is a linear increase in the critical resolved shear stress (CRSS) of NWs at the yield point under increased twin boundary density or decreased twin boundary spacing. For instance, the CRSS value obtained for SC NWs can introduce coherent twin boundaries during the growth of NWs, resulting in either detrimental or beneficial effects on the mechanical property (yield stress) [54,61]. 

The crystalline structure of Ag NWs plays an important role in the mechanical strength. For instance, the Young’s modulus of Ag NWs is different for SC and fivefold-twinned (FT) NWs depending on the elastic anisotropy. Typically, materials with large diameters show an increased Young’s modulus, whereas both decreasing and increasing Young’s moduli were observed for materials with smaller diameters. The SC Ag NWs show an increased Young’s modulus at decreased diameter. There is a difference in the Young’s moduli for FT and SC NWs because of the compatibility constraint imposed by the structure. Different from SC NWs, the Young’s modulus of FT NWs does not change with a decrease in the diameter, suggesting an independent mechanical strength behavior of FT NWs. Furthermore, FT Ag NWs show more enhanced yield strength under tensile load compared with their SC counterparts. However, under compression, FT Ag NW shows a lower yield stress compared with the corresponding SC Ag NW [54,61]. 

### 3.2. Mechano-Electric Behavior of Ag NW Networks 

The mechano-electric properties of Ag NW networks should be critically evaluated in designing and fabricating flexible electronics with high reliability. Considering the direct mechanical deformation of NWs on the devices, it is crucial to prevent the fatigue failure of Ag NWs after repeated bending deformations. Typically, the mechano-electric testing of conventional macroscale materials, such as ITO, can be performed by tensile and bending testers. However, in contrast to macroscale materials, there is no standard test method or equipment for the mechano-electric characterization NW networks. With respect to the mechanical characterization of NW networks, bending fatigue testers have been developed for the evaluation of the long-term mechanical reliability of Ag NW-based electrodes under various deformation environments. In this section, we discuss the mechano-electric behavior of Ag NWs under repeated bending cycles exposed to various conditions or environments [48,49,50,51,52,53,54,55,56,57,58,59,60,61,62,63].

#### 3.2.1. Dynamic Mechano-Electric Property of Ag NW Networks 

As Ag NW-based flexible devices are continuously exposed to the repeated external stresses, Ag NW networks can exhibit increased resistance after multiple cycles of mechanical deformations [49,63]. Under bending, Ag NW networks are extended by the application of tensile stress. In this context, it is worthwhile to comprehensively understand the fatigue behavior of Ag NWs for the assessment of their durability as safety-critical structural components, as well as for guiding the design, fabrication, and optimization of NW-based devices [49,63]. For the analysis of the long-term mechano-electric reliability of Ag NW networks, it is required to understand their dynamic (time-dependent) mechanical properties (or bending fatigue behavior). In this regard, the electrical properties of Ag NWs should be correlated with their deformation behavior under compression and tensile stress environments using a bending fatigue tester. 

As the mechanical resilience of a device can be evaluated under repeated deformation, the flexibility of NWs is characterized by plotting the resilience as a function of mechanical bending cycles in which the mechanical strain is controlled by changing the bending radius (Figure 11a). As shown in Figure 11b, the bending fatigue test for Ag NW networks can be carried out by setting the bending radius (*R*), which is defined as half of the gap of the bent substrate [64,65]. The bending fatigue resilience is defined by measuring the fractional change in resistance (*ΔR/R_0_* = *(R − R_0_)/R_0_*). It should be noted that a fatigue test for an Ag NW network is difficult to perform owing to the time-consuming experiment requiring a sufficient number of bending cycles (> ~1000). In addition, the microstructural analysis of an Ag NW network can be performed by associating the microstructure of the Ag NWs with the number of bending cycles. The position of an Ag NW on the substrate determines the stress applied to the Ag NW. When the NW is placed on the upper end of the neutral plane, tensile stress is applied, whereas compressive stress is applied when it is placed on the lower end [1,2,3,4,5,49,63,64,65]. 

The bending fatigue behavior of Ag NW networks is closely associated with their cracking life. Specifically, a crack induced by bending fatigue intensifies the bending stress, leading to the propagation of fatigue cracks under increased bending cycles. Similar to conventional defects (e.g., dislocations and cracks), defects on Ag NWs can act as local stress concentrators and serve as sources of crack nucleation and propagation. In addition, Ag NW networks exhibit structure (location)-dependent behavior under applied stress on the NW. In this respect, Ag NWs exhibit superior bending fatigue resistance to that of their bulk counterpart under typical bending fatigue test conditions; the facilitated atomic mobility on the surface of NWs with a large surface-to-volume ratio enables the reduced fracture of NWs under extended cyclic deformations. Theoretically, the defect-free nature (or lack of crack nucleation) of intact NWs enables infinite fatigue life [50,51,52]. As will be discussed below, there are several parameters affecting to the mechanical strength of Ag NWs under repeated bending cycles. 

(1) Density of Ag NW networks

Reportedly, the density of Ag NW networks can critically affect their physical properties and deformation behavior under cyclic bending fatigue testing [49,50,51,52,53,63]. Different from the channel crack-induced fatigue failure of metal thin films, the failure mode of Ag NWs network depends on their density, because of the frequent occurrence of the failure (local failure) of Ag NW networks at junctions. The parameter *n* represents the number of unit squares in a line; thus, a higher ***n*** value can be considered as a higher density of the Ag NW network. As revealed in Figure 12, the mechano-electric reliability of Ag NWs can be closely associated with their network density, suggesting the importance of the optimized geometrical structure of Ag NW networks toward various electronic devices. A simulation study showed that the failure of Ag NWs was more pronounced for the densest NW networks, owing to the more confined geometry. Specifically, a NW network with a confined geometry and higher density less able to accommodate an applied bending strain by stretching. That is, denser Ag NW networks exhibit an increased resistance compared to those of lower-density networks [49,50,51,52,63,66].

(2) Size and geometric effect 

Typically, nanostructured materials exhibit an increased fatigue life in the high-cycle regime after the bending fatigue test, indicating the increased yield strength and reduced ductility [49,50,51,52]. Such an enhanced mechanical reliability of Ag NWs can be attributed to the size effect, whereby multiple dislocations of Ag NWs become increasingly difficult to create in the submicrometer regime. 

The electrical percolation of Ag NW networks also depends on their dimensions, resulting in variable mechanical reliability (Figure 13). Reportedly, Ag NW networks with narrower widths show drastic (Figure 13a) increases in resistance during the bending cycles [49,50,51,52]. An Ag NW with a larger width contains more electrical percolation pathways than narrow strips, leading to enhanced reliability of the network. With respect to the directionality of an Ag NW network, it does not heavily affect the failure behaviors, as most of NW network has some form of random distribution and exhibits no directionality in resistance (Figure 13b) [49,50,51,52]. 

(3) Environmental effect 

The environmental conditions also affect the mechanical reliability of Ag NW network-based devices. Specifically, although Ag shows superior chemical stability to other TCE candidates, such as Cu or Ni, Ag NW-based TCEs are still sensitive when exposed to air, owing to their high surface area. The reduced cross-sectional area of NW under oxidation results in a drastically enhanced resistance of the electrode. Then, as the stress is concentrated on the oxidizing region under bending, the crack probability increases with NW degradation. The mechanical properties of Ag NW network-based devices can also be deteriorated by their oxidation and sulfidation as well as by the electro-migration of Ag atoms [67]. It should be noted that, in practical applications, device aging is not directly or solely correlated to the deterioration of the material (Ag NW) by the environment but depends more on the overall architecture of the device owing to their complex ageing mechanisms [67]. 

#### 3.2.2. Enhanced Mechano-Electric Property 

The network structure of Ag NWs suggests superior mechanical reliability under bending fatigue to that of un-networked NWs owing to the effective accommodation of the bending strain by the network stretching and the limited accruement of dislocations within the NW network [49,50,51,52,68]. Nevertheless, the mechano-electric performance of Ag NWs is insufficient for application to commercial devices. It is required to improve the mechano-electric stability of Ag NW networks by properly addressing the current challenges facing the industrial integration of Ag NWs into devices. In this context, several protocols have been developed for the preparation of Ag NW-based TCEs with enhanced mechanical flexibility, such as welding, compositing, or deposition of extra layers onto Ag NW. 

(1) Modification of NW network

(a) Annealing/welding of NW network

Annealing/welding can effectively enhance the mechanical stability of TCEs, allowing them to withstand higher mechanical stress without sacrificing their electrical properties by reducing the contact resistances between Ag NWs. As the conductivity of Ag NW networks is mainly influenced by their surface and grain boundary scattering, the conduction mechanism changes from tunneling to free electron conduction at increased Ag NW concentration. 

In this context, the electrical conductivity of Ag NWs can be effectively enhanced by local sintering of NW junctions through various annealing processes, including thermal annealing, light irradiation, mechanical pressing, plasma treatment, extra coating, cold welding, laser irradiation, humidity, and chemical treatment [68,69,70,71,72]. Specifically, as demonstrated by Giusti and Langley et al., the thermal annealing of an Ag NW network forms fused-in junctions among the NWs, leading to significantly enhanced flexibility of TCEs with retained electrical conductivity after long-term (500 K) cycles under the strain of 1% [44,45,69]. As demonstrated by Li and Park et al., the welding of Ag NWs with intense light (e.g., flashlight) irradiation (4.6–10.3 J/cm^2^ per pulse), high-power tungsten–halogen lamp (30 W/cm^2^) or plasma treatment are also suggested to enhance the mechano-electric reliability (or maintain the electrical resistance) of Ag NWs for long-term bending cycles (>10,000) [35,73,74,75]. As for welding by an extra coating layer (ITO) by sputtering, a welded Ag NW network exhibits maintained electrical resistance after 10,000 bending cycles [35,76,77]. 

Recently, there have been studies on the effect of the welding process on the mechano-electric property of Ag NWs. In typical thermal annealing processes, the NW junctions were welded with increased temperature, leading to a reduced percolation threshold (enhanced conductivity). However, as shown in Figure 14a, cold-welding processes, including capillary-force-induced or mechanical welding, create self-limited welding of the interwire junctions of Ag NW networks, yielding improved fatigue characteristics [35,64]. The welded Ag NW network can be formed through a mechanical joint by applying the mechanical stress (or bending stress) on the NW network [64,75]. Very interestingly, as shown in Figure 14b, an enhanced mechano-electric reliability was observed for an un-annealed Ag NW network [64]. 

(b) Regulation of the alignment of the NW network 

Regulating the alignment of Ag NW networks and their percolation behavior can further enhance the electrical conductivity of the networks. Figure 15 compares different NW electrical percolation networks (∼22 Ω/sq.) of aligned and random Ag NW networks [1,2,3,4,5,78,79]. Specifically, an aligned Ag NW network precisely controlled through capillary printing using a nanopatterned PDMS stamp can have a lower NW density compared to that of a random Ag NW network, owing to the lower percolation threshold. As shown in Figure 15a, the aligned Ag NW films showed significantly reduced sheet resistance at the same NW surface density. The reduced percolation thresholds of the aligned Ag NW networks allow higher optical transmittance (***T***) than those of random Ag NW films at similar ***R_s_***. Figure 15b compares *T* and the haze factor of aligned and random Ag NW networks with similar ***R_s_*** values (~22 Ω/sq.). The aligned Ag NW networks exhibited ∼3% higher ***T*** and 2.4 times lower haze values at 550 nm wavelength, compared with the random Ag NW networks. These enhanced optical properties of the aligned Ag NW networks can be attributed to the decreased light scattering from the reduced NW surface density. The mechanical stability of flexible polymer light-emitting diodes (PLEDs) with aligned Ag NWs (Figure 15c) showed 80% retention of the initial luminance over 300 bending cycles, whereas ITO-based flexible PLEDs showed rapid decreases in luminance. Notably, PLEDs using aligned Ag NW electrodes showed a 30% enhanced maximum luminance (33,068 cd/m^2^) and a higher luminance efficiency (14.25 cd/A) compared with those using random Ag NW networks [1,2,3,4,5,78,79]. Moreover, Ag NWs networks can be regulated by optimizing the scale of the Ag NW network to enhance the device efficiency [78,79]. For instance, the network structure of shorter/smaller Ag NWs (10 mm/40 nm) deposited on the percolation voids of relatively longer/larger diameter Ag NWs (100 mm/100 nm) can enhance the OLED efficiency owing to the dual-scale metal NW network [1,2,3,4,5,78,79].

(2) Hybridization 

The mechano-electric properties of Ag NW percolation network-based TCEs can be further enhanced in combination with other materials or stacking with appropriate functional layers through various hybridization techniques [80,81,82,83,84,85,86,87,88,89,90]. 

(a) Ag NW–oxide hybrid

The Ag NW–oxide hybrid TCE exhibits superior chemical and mechanical characteristics (stress–strain behavior and fatigue characteristics) to those of Ag NW-based TCEs. Such Ag NW–oxide hybrids can be constructed through the compositing of Ag NWs with various oxides, including ITO, ZnO, TiO_2_, Al_2_O_3_, and graphene oxide (GO), yielding enhanced mechanical properties. 

For instance, the hybrids of Ag NW–reduced graphene (rGO) (Figure 16a), Ag NW–TiO_2_ nanosheet (NS) (Figure 16b), and Ag NW–ZnO (Figure 16c,d) exhibit the improved fatigue behavior owing to the limited crack formation of Ag NWs after application of the secondary nanostructures (GO NS, TiO_2_ NS, and ZnO film). More specifically, the application of GO NS can effectively improve the chemical and mechanical reliability of an Ag NW network owing to its low water and gas permeability and high chemical resistance, making it suitable as a protective film for flexible devices, as well as enhancing the mechanical reliability of the Ag NWs. In addition, GO NS can solder Ag the connections in the NW network, leading to a clearly reduced inter-NW contact resistance without high-force pressing, or a heat treatment processes [80,81,82,83,84]. With respect to metal oxides (Figure 16b–d), despite the poor conductivity of the metal oxide layer (ZnO, TiO_2_, Al_2_O_3_), they can reduce the contact resistance of Ag NW networks, thereby decreasing the total resistance. Specifically, Ag NW failures by Joule heating from electrical current can be prevented by applying a conformal layer of ZnO wrapped around the Ag NW (ZnO coated on Ag NW). As a result, OLED devices based on a ZnO–Ag NW hybrid demonstrate longer lifetimes compared to ITO/Ag NW-based OLED devices [85,86,87,88].

(b) Ag NW–polymer hybrid 

There are great challenges facing the development of highly stretchable conducting architectures based on Ag NW–polymer composites to improve their deformability, electromechanical stability, and fatigue-resistance [65,89,90,91,92,93,94]. As Ag NW TCEs are required to withstand external mechanical forces (tensile, compressive, and shear forces), both the Ag NW film and the polymer should be stretchable, and the latter acts as a supporting layer for the Ag NW network. Typically, highly stretchable composites are prepared by assembling an Ag NW network into a sponge-like polymer skeleton, such as PDMS, poly(urethane acrylate) (PUA), or polyimide [89,90,91,92,93,94]. As a result, the mechanical stability of stretchable Ag NW conductors can be increased by compositing Ag NWs with a polymer as a supporting substrate. For instance, as shown in Figure 17a,b, a hybrid film (Ag NW/PEDOT:PSS) displays superior bending and taping stability compared to those of bare Ag NWs, indicating enhanced mechano-electric stability [89,90]. However, the structural design is also needed to ensure the electrical conductivity of the film under strain. For instance, as shown in Figure 17c (top: schematic, bottom: stretched devices being twisted/crumpled and strained), if Ag NW networks are pre-strained to form a wrinkled structured film or have a wavy structure by compression of floating film onto PDMS, the TCEs show significantly enhanced mechano-electric performance (or stretchability) [1,2,3,4,5,91,92]. 

(c) Ag NW–conductive layer hybrid 

An Ag NW–conductive layer hybrid is beneficial to compensate the mechanical failure of NWs with the aid of an auxiliary conductive layer. For instance, if Ag NWs are electrically disconnected or have poor conductivity, the auxiliary conductive layer (ITO or graphene) becomes the main pathway for electronic transport [95]. An Ag NW/graphene hybrid shows a higher conductivity than individual Ag NWs and graphene in parallel, suggesting the potentially higher mechanical endurance of the hybrid electrode against bending fatigue. As shown in Figure 18a, an Ag NW/graphene hybrid film exhibits an increase in resistance of 25%, which is much less than that (*ΔR/R*_0_ > 140%) shown by the Ag NW film after cyclic bending tests under 6.5% strain, owing to new current paths formed by the graphene layer underneath the Ag NWs. Although some Ag NWs are damaged by the bending fatigue, the graphene can effectively suppress the resistance increase of the electrode under high bending strain. In addition, in contrast to the result of the Ag NW/graphene sheet hybrid film that showed a sudden increase in resistance, the Ag NW/graphene mesh hybrid film exhibited a gradual resistance increase without any abrupt increase, even after 100,000 bending cycles. The superior mechano-electrical stability of Ag NW/graphene mesh film is further corroborated by displaying its retained morphology after cyclic bending (Figure 18b) [83]. 

(3) Enhancing the compatibility between Ag NW and substrate

As Ag NW films require supporting substrates owing to their limited self-supporting characteristics, it is required to improve the compatibility of Ag NWs with the substrates and other coating layers to ensure the high mechano-electric reliability of Ag NW films. For instance, good adhesion between Ag NWs and the substrates is crucial to maintain the mechanical stability of TCEs under repeated bending cycles. In addition, poor adhesion of bare Ag NW films to most substrates has necessitated enhancing the adhesion between Ag NWs and the substrates by the supplemental processes and compositing with other materials. Lee et al. demonstrated enhanced adhesion between Ag NWs and PET substrates through the lamination of Ag NWs on the substrate at 120 °C, leading to stronger contact between the NWs and substrate [14,35]. The intense-pulse-light (IPL) method was also employed to enhance the adhesion between Ag NWs and the substrate by briefly heating the NWs and substrates at high temperature, thus increasing the contact between the materials [35,65]. 

As non-heating approaches, external and strong conformal pressures are applied to Ag NW electrodes thereby improving the adhesion of Ag NW to the substrate (e.g., PET) by coating an extra layer or directly compositing the Ag NWs. For instance, an extra layer such as a metal oxide (ZnO, TiO_2_) or graphene can be coated onto Ag NWs through atomic layer deposition or sol–gel methods, leading to enhanced adhesion of the Ag NWs on the polymer substrates [14,30,35,76,77,80,86]. As demonstrated by Nam et al., the adhesion of Ag NWs was greatly improved by the embedding of Ag NWs in polymers, such as Norland Optical Adhesive, chitosan, alginate, and polyvinyl alcohol (PVA) [35,96]. 

## 4. Applications of Ag NW Networks with Mechano-Electric Properties 

As discussed in the previous section, there have been many Ag NW-based devices developed for stretchable and flexible electronics (e.g., stretchable displays, high-frequency antennas, artificial muscles, and skin sensors) [1,2,3,4,5,8,96,97,98,99,100,101,102,103,104,105,106,107,108,109,110,111]. In this section, we introduce Ag NW network-based flexible optoelectronic and electronic devices, emphasizing their performance as a function of their mechano-electric properties.

### 4.1. Optoelectronic Devices 

Ag NW-based TCEs for various optoelectronic devices (displays, multifunctional sensors, touchscreens, etc.) have been intensively studied, owing to the good electro-optical properties and mechano-electric stability of Ag NW networks. In this part, we discuss the mechano-electric performance of Ag NW-based devices under dynamic motion during extended operation periods. 

#### 4.1.1. Transparent Conductive Electrode

As noted, the mechanical properties (e.g., adhesion to substrates and flexibility) of Ag NW-based TCEs play an important role in determining the compatibility of device fabrication and scalability (R2R processing). The adhesion property of Ag NW films was evaluated by a peeling test with Scotch tape and the resistance of the Ag NW films was recorded before and after the taping process (Figure 19). Ag NW and Ag NW–TiO_2_-based electrodes prepared under different thermal conditions (200–350 °C) exhibit different morphologies (Figure 19a (Ag NW network (300 °C)) and 19b (Ag NW−sol−gel TiO_2_ film (300 °C)) and resistance changes (Figure 19c), depending on the annealing conditions. Specifically, as shown in Figure 19c, Ag NW–TiO_2_ shows a retained NW structure, even after annealing at 300 °C, whereas the network structure of bare Ag NWs is destroyed at moderate temperature (300 °C). After the peel-off test using Scotch tape, the sheet resistance of Ag NW−sol−gel TiO_2_ did not change, whereas the bare Ag NW network showed increased sheet resistance beyond the measurement limit (120 MΩ/sq.). This retained conductivity of Ag NW–sol–gel TiO_2_ is attributable to the function of the sol−gel TiO_2_ as a mold to fix the Ag NWs to the substrates. More importantly, Figure 19d shows that the composite exhibits superior performance to that of sputtered ITO on PET substrates after 500 bending cycles at 5R bending radius [87]. 

#### 4.1.2. Flexible Optoelectronic Device 

(1) Flexible Organic Light-Emitting Diode (OLED) 

The performance of Ag NW-based electronic devices with very thin transparent active layers (organic compounds, metal oxides, etc.) is further enhanced by reduced surface roughness and work function regulation, which lead to enhanced charge carrier injection or collection in the devices [87,96,97,98,99]. More importantly, such modifications could improve the mechano-electric properties of the devices [98]. For instance, as shown in Figure 20, Ag NW meshes on PET were prepared without (Figure 20a) and with (Figure 20b) roll pressure treatment to compare the effect of roll pressure on the electrical performance of the Ag NW mesh. As shown in Figure 20c, bending the Ag NW meshes showed no conductivity degradation before and after bending at different radii of curvature (50–0.625 mm). Repeated bending at 1R also showed no increase in the sheet resistance, demonstrating good adherence and no signs of delamination of the NW mesh array. This result indicates the reduced sheet resistance of Ag NW mesh films between the individual Ag NWs by roll pressing. In the Ag NW-incorporated optoelectronic devices, Ag NW mesh electrodes exhibit comparable device yield with that of ITO-based devices, owing to the reduced surface roughness of the electrodes by roll pressing [98]. 

There have been many flexible optoelectronic devices (e.g., stretchable PLED)) developed based on Ag NW networks and polymers [1,2,3,4,5,97,98,99,100,101,102,103]. Typically, flexible OLEDs can be constructed by the deposition of OLED components (e.g., PEDOT: PSS/NPB/Alq_3_/LiF/Al) on an Ag NW-based electrode, where an Ag NW–polymer composite-based TCE can be prepared by utilizing elastic polymers (e.g., PVA and PUA). For instance, Figure 21A(a) shows a stretchable electroluminescent (EL) device with a sandwich structure of TPU-Ag NW/EL layer/TPU-Ag NW (composite electrodes of Ag NW and high-*k* thermoplastic polyurethane (TPU)). The as-prepared stretchable EL device (Figure 21A(b)) shows an enhanced efficiency as well as uniform light emission in the bent, stretched, and twisted states, owing to the significantly reduced surface roughness. In addition, as shown in Figure 21A(c), the annealed Ag NW/TPU electrode shows a much smaller resistance increase than that of the as-deposited electrode under uniaxial stretching to 100% strain. As for the mechano-electric performance during repeated bending cycles, different levels of resistance change are observed during 1000 stretching−relaxing cycles to 50% strain (Figure 21A(d)). Such excellent mechano-electric stability can be attributed to the strong physical adherence of Ag NWs to the elastomeric substrate, which can limit the junction damage and inter-NW sliding for improved stretchability [97]. Similarly, an Ag NW–PVA composite electrode exhibits excellent mechanical stability (adhesion, friction, and bending stability) because of the strong anchoring of Ag NWs to the PVA matrix at 120% strain [100]. Figure 21B displays the structure (Figure 21B(a)) and a photograph (Figure 21B(c)) of a stretchable PLED based on a GO–Ag NW/PUA composite and of a semi-transparent stretchable PLED. In addition, GO–Ag NW/PUA exhibits negligible resistance changes after a long cyclic bending fatigue test (Figure 21B(b)), leading to retained PLED performance even in the stretched state (130%) (Figure 21B(d)) [102]. Figure 21C shows the electronic and mechano-electric performance of an OLED of alginate (Alg)-based composite film, NaAlg(CaCl_2_)/Ag NW composite film, and ITO anodes. Note that, as shown in Figure 21C(a), the Ag NW composite electrode exhibits superior bending stability to that of the controls (ITO and bare Ag NW). In addition, the Ag NW composite electrode exhibits superior current efficiency, depending on the luminance, to that of the ITO electrode (Figure 21C(b), owing to the formation of the composite structure [101,103].

### 4.2. Electronic Devices

Flexible electronic devices based on Ag NW networks require both desired electrical and mechanical properties. In this section, we introduce a flexible heater and electromagnetic interference shielding, as two examples out of many Ag NW-based flexible electronic devices. 

#### 4.2.1. Wearable Electronic Devices 

Wearable electronic devices (e.g., clothing-integrated sensors, heaters, and portable devices) require electrically conductive materials with mechanical flexibility. The candidate materials for wearable devices (e.g., e-textiles and polymer composites) should be mechanically flexible without significant degradation after repeated bending cycles. 

First, e-textiles can be prepared by coating the surface of fibers (nylon, polyester, and cotton) with Ag NWs, leading to mechanically flexible and electrically robust composites after repeated bending. As shown in Figure 22A, e-textile can be prepared by uniform coating of NWs on the fiber (nylon thread). The resistance variations under repeated bending for commercial conductive thread and a NW-coated nylon thread with the same initial resistance are plotted in Figure 22B(a), exhibiting that the resistance of the NW-coated thread only increased by 14% after 200 bending cycles at 6R bending radius, whereas the resistance of the commercial thread increased by a factor of more than four over the same period [105]. The enhanced mechanical stability of Ag NW-coated thread can be attributed to the unique network structure of Ag NWs, with much more flexibility and endurance of higher elastic strains. In addition, the mechanical forces caused by bending improve the connections of the NW junctions, thereby reducing the junction resistances. As plotted in Figure 22B(b), similarly to the result of commercial threads, the retained resistance of the Ag NW-coated nylon thread after repeated washings with detergents indicates the robustness of the coating on the thread and good adhesion of Ag NWs to the nylon threads [105]. 

#### 4.2.2. Flexible Heater

A flexible heater, especially a stretchable transparent heater, is another application of Ag NW-based TCEs, owing to the high electrical and thermal conductivity of Ag NWs. Typically, a transparent thermal heater can be prepared mixing Ag NWs with an optimal amount of nonconducting materials to induce Joule heating [1,2,3,4,5,6,7,8,43,44,45]. As shown in Figure 23a, Lee et al. demonstrated a flexible heater based on an Ag NW-coated PET substrate, exhibiting mechanical stability, high optical transmittance and conductivity, as the value of Δ***R_s_*** remained below 0.1% after the bending tests [1,2,3,4,5,106]. Such a negligible change in the sheet resistance indicates the excellent mechanical robustness of the Ag NW film fabricated by supersonic cold spraying. Figure 23b shows the stretching of an Ag NW-based heater on a flexible substrate (eco-flex), where the heater film can be stretched up to 400% without a noticeable degradation in the heating property. It should be noted that the reciprocal relation between the transparency and conductivity of the Ag NW film limited the concentration of Ag NWs present in the film, leading to limited stretching of the heater beyond 400%. Furthermore, a patterned Ag NW-coated flexible PET heater could be prepared using a stainless-steel mask. This indicates that the heat-releasing capability of patterned Ag NWs was retained over the structure. In a similar study, Hong et al. introduced a stretchable thermal heater made of a Ag NW-deposited PDMS film by exploiting the swelling phenomenon of PDMS [1,2,3,4,5,106,107,108,109,110]. The resultant stretchable heater has an effectively embedded Ag NW percolation network within the PDMS matrix and good thermal stability at strain of 60%. Along with device fabrication, Sorel et al. reported the characteristic changes in an Ag NW-based transparent thermal heater in the bulk-like and percolative regimes by calculating a percolative figure of merit of the Ag NW percolation network [1,2,3,4,5,106,107,108,109,110]. The percolation network model of Ag NWs provides critical considerations, including the electrical conductivity, optical transparency, and heat dissipation for high-efficiency Joule heaters [1,2,3,4,5,106,107,108,109,110]. The Ag NWs coated on the eco-flex substrate showed effective electro-thermal performance in the rapid on/off test and good mechano-electrical stretchability up to 400%. 

#### 4.2.3. Electromagnetic Interference Shielding

A highly stretchable and transparent electromagnetic interference shielding (STEMIS) layer for wearable electronic devices was demonstrated with an Ag NW network on a PDMS substrate (Figure 24a). As shown in Figure 24b, the EMI shielding effectiveness of the device was measured under various strain conditions by installing an X band waveguide between the moving stage and placing a STEMIS film on the waveguide (Figure 24b). The STEMIS layer with different Ag NW areal densities exhibits a high electromagnetic wave shielding effectiveness under a high strain condition (Figure 24c). Specifically, when the Ag NW density of the STEMIS is higher than 333 mg/m^2^, the EMI shielding effectiveness is maintained at 20 dB or higher, even at a large strain of 50%, indicating stable EMI shielding effectiveness under mechanical strain. As shown in Figure 24d, despite the decreased EMI shielding effectiveness at increased sheet resistance, the degradation of EMI shielding effectiveness is slowed down with a decrease in the Ag NW density owing to the increased sheet resistance by stretching. Considering the increase in the EMI shielding effectiveness at low Ag NW density, this unique phenomenon is attributable to the effective shielding of the incoming EM wave by isolated networks owing to the highly conductive metal network [111]. 

## 5. Summary and Concluding Remarks 

This review presents the synthetic methods, properties, and applications of Ag NW networks by focusing on their mechano-electric properties for applications to various flexible devices and summarizing notable findings and cases from the recent literature. 

First, we discuss Ag NW network-based electrodes prepared through various coating processes and their chemical, electrical, and optical properties. The deposition processes of Ag NW networks on certain substrates have been addressed, from solution-based coating processes (drop casting, spray coating, spin coating, etc.) to commercial processes (slot-die and R2R coating). The electrical characteristics of Ag NW networks are also discussed by focusing on the electrical properties governed by percolation and the electrical contacts of the networks. 

Second, the mechano-electric properties of Ag NW networks are reviewed by describing individual properties (electrical and mechanical properties) of NW networks with dynamic motion under cyclic loading, as well as their combined properties. Whereas the mechanical characteristics such as flexibility and stretchability of Ag NW networks are primarily governed by the mechanical robustness of the individual NWs, the mechano-electric properties of NW networks are affected by both the electrical percolation and connections of the networks. The improved mechano-electric properties of Ag NW are also discussed by presenting Ag NW network-based flexible electrodes prepared through various approaches, including post-treatment and hybridization. 

Third, various device applications of Ag NW network-based flexible electrodes are discussed. Specifically, electronic and opto-electronic devices are discussed by introducing the basic strategies, applications, and challenges of each device. 

Despite the many advantageous features (good electro-optical performance, stretchability, and long-term mechanical stability) and expected significant role of Ag NW electrodes in the wearable device industry, there is still much progress to be made to achieve the future commercialization of small, flexible, and mechanically robust electronic devices. We believe that this review will not only serve as a design guide for fabricating Ag NW network-based flexible electronic devices with high mechano-electric reliability, but will also be helpful to fundamentally understand the various mechanical and electrical properties of Ag NWs.

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
