# Peer review of "Silver Nanowire Networks: Mechano-Electric Properties and Applications"

_materials, 2019, doi:10.3390/ma12162526_

Round 1

Reviewer 1 Report

Dear Authors,

Unfortunately, I could not recognize that you have advanced the state-of-the-art on Silver Nanowire reviews. What you summarize is better presented in previous review articles that date back to 2013, e.g. by Langley et al.:

https://iopscience.iop.org/article/10.1088/0957-4484/24/45/452001

Further, I do not recommend using pixel graphics (see page 12-14).

For a review article, you should think how to visualize the key topics for AgNWs in 1-2 own figures rather than copying tables from previous review articles or just inserting numerous figures from publications (of course you also need to do that but not to the degree that you did). Also, you should work on improving the structuring and in particular the abstract section that reads like the one of an original research paper and not the one of a review.

Therefore, I recommend to reject your submission with an eventual recommendation for resubmission.

Author Response

Response to Reviewers’ comments

02-Jul-2019

Journal: Materials (ISSN 1996-1944)

Manuscript ID: materials-538840

Type: Review

Title: " Silver Nanowire Networks: Mechano-Electric Properties and Applications"

Author(s): Hiesang Sohn, Chulhwan Park, Jong-Min Oh, Sang Wook Kang, Mi-Jeong Kim

Dear Reviewers,

Thank you for your careful review and valuable comments. We carried out major revision of the paper by considering all the points raised by reviewers and editor. After this process, we believe our paper has been significantly improved.

In this “Author’s response to reviewers’ comments”, we answered/addressed all the points raised in reviewers’ comments. Our responses to each point brought up are given in blue. We thoroughly revised the paper while revised parts are highlighted in yellow.

Reviewer Comments to Author:

Reviewer #1

Comment 1:

Comments: Dear Authors, Unfortunately, I could not recognize that you have advanced the state-of-the-art on Silver Nanowire reviews. What you summarize is better presented in previous review articles that date back to 2013, e.g. by Langley et al. https://iopscience.iop.org/article/10.1088/0957-4484/24/45/452001 Further, I do not recommend using pixel graphics (see page 12-14). For a review article, you should think how to visualize the key topics for AgNWs in 1-2 own figures rather than copying tables from previous review articles or just inserting numerous figures from publications (of course you also need to do that but not to the degree that you did). Also, you should work on improving the structuring and in particular the abstract section that reads like the one of an original research paper and not the one of a review. Therefore, I recommend to reject your submission with an eventual recommendation for resubmission.:

Answer to comment 1:

Thank you for your time and effort on my paper with your comment. As reviewer pointed out, our paper requires revision by correcting some parts and providing with more original data (Figures and Table). Nevertheless, we believe our article contribute to the field of flexible electronic devices since it focuses on the preparation, properties and applications of Ag NW network. More specifically, this review focuses on the mechano-electric properties of Ag NW networks for the applications towards various flexible electronic devices by summarizing notable findings and cases in the recent researches.

According to reviewer’s comment, we thoroughly revised the manuscript by performing follows. We believe the quality of our manuscript has been much improved after careful revision, being ready for published in “Materials

Restructuring the text of manuscript: We made manuscript more logically by restructuring the order of subsections. In addition, we streamlined the manuscript by removing redundant or unnecessary contents.

Updating data (Figures and Table): We insert new original figures and replace table with new one. In addition, we updated Figures by re-drawing figure clearer and making captions more readable/recognizable.

Rewriting abstract: As reviewer pointed out, our abstract is a little bit long and separated paragraphs seem to be unusual. We revised the abstract by shortening the length of abstract and merging two paragraphs. In addition, we made abstract text more logically intercorrelated

English editing: We revised the manuscript for grammar correction by professional English and academic editing services (e.g., Editage). Here we provide with the certificate of English Editing as shown in below.

Reviewer 2 Report

Please, find comments in attachment.

Author Response

Response to Reviewers’ comments

02-Jul-2019

Journal: Materials (ISSN 1996-1944)

Manuscript ID: materials-538840

Type: Review

Title: " Silver Nanowire Networks: Mechano-Electric Properties and Applications"

Author(s): Hiesang Sohn, Chulhwan Park, Jong-Min Oh, Sang Wook Kang, Mi-Jeong Kim

Dear Reviewers,

Thank you for your careful review and valuable comments. We carried out major revision of the paper by considering all the points raised by reviewers and editor. After this process, we believe our paper has been significantly improved.

In this “Author’s response to reviewers’ comments”, we answered/addressed all the points raised in reviewers’ comments. Our responses to each point brought up are given in blue. We thoroughly revised the paper while revised parts are highlighted in yellow.

 Reviewer #2

Comment 1:

The paper presents a good review in the topic of silver nanowire networks, with very detailed overview of fabrication, structural and electronic properties. Moreover, this paper shows a good overview of applications of such structures. The paper combines the information that can be useful for researchers in different areas. In general, paper is well structured and well written, however I have few comments, that can help to improve the manuscript.

Answer to comment 1:

Thank you for your comment. As reviewer pointed out, our paper requires revisions in terms of figures and text. According to reviewer’s comment, we thoroughly revised the manuscript by performing follows.

1) Restructuring the text of manuscript: We made manuscript more logically by restructuring the order of subsections. In addition, we streamlined the manuscript by removing redundant or unnecessary contents.

2) Updating data (Figures and Table): We inserted new original figures and replace table with new one. In addition, we updated Figures by re-drawing figure clearer and making captions more readable/recognizable.

3) Rewriting abstract: As reviewer pointed out, our abstract is a little bit long and separated paragraphs seem to be unusual. We revised the abstract by shortening the length of abstract and merging two paragraphs. In addition, we made abstract text more logically intercorrelated.

English editing: We revised the manuscript for grammar correction by professional English and academic editing services (e.g., Editage).

Comment 2:

1) Abstract is a little bit long, and the separation it into two blocks is confusing

Answer to comment 2:

Thank you for your comment. As reviewer pointed out, the abstract is a little bit long and separated paragraphs seem to be unusual. According to reviewer’s comment, we revised the manuscript by shortening the length of abstract and making single paragraph by merging two paragraphs.

Comment 3:

2) On the page 2, lines 57-58: The sentence has “to their” three times in a row, that makes the sentence confusing.

Answer to comment 3:

Thank you for your comment. As reviewer pointed out, the repeated expression (to their) seems to make the confused sentence. According to reviewer’s comment, we made the text clearer by removing the redundant words (to their) and by streamlining the full sentences as shown below.

Previous sentence: In this context, there have been keen interests in nanostructured materials because of their superior physical properties to their bulk counterparts relevant to their potential application towards electronic and optoelectronic devices. At the same time, the needs for new materials for the flexible electrodes are emerging to overcome the shortcomings of conventional electrodes to be applied in flexible/wearable electronics [1-4]. 

Revised sentence: In this context, there has been much interest in the application of nanostructured materials to flexible electronic and optoelectronic devices by convenient processes to overcome the shortcomings of conventional electrode materials (e.g., ITO and other oxides) [1-4].

As we may have similar issues in other parts in the manuscript, we revised the manuscript for grammar correction by professional English and academic editing services (e.g., Editage). Here we provide with the certificate of English Editing as shown in below.

Comment 4:

3) Section 2.1.1, Figure 2. It is clear from the figure 2(b) and (c) that substrate, where NWs re depositing is influence a lot on the morphology of the network, maybe authors can say a few words about this influence in the section 2.1.1. This can be useful for other researchers in this field.

Answer to comment 4:

Thank you for your comment. As reviewer pointed out, it is meaningful to describe the morphology of the network prepared through repeated depositions of NWs by drop casting for other researchers. Note that, despite the simple deposition process of drop-casting, it is known that Ag NW films formed by drop-casting can display non-uniform surface morphologies. According to reviewer’s comment, we revised the manuscript by more describing the morphologies and properties (electrical and optical properties) of multiply deposited Ag NW films by drop-casting method.

Comment 5:

4) Figure 1, page 2. Authors can make titles for each illustrated method more clear and more correlated with next sections of this review paper.

Answer to comment 5:

Thank you for your comment. As reviewer pointed out, it can be useful to put the title of illustrated method to making more clear and correlated form. According to reviewer’s comment, we revised the manuscript by more specifically and correlatedly describing the titles of fabrication methods in the Figure 1.

Comment 6:

5) Figure 5(B) page 8, captions for images are very small, difficult to read.

Answer to comment 6:

Thank you for your comment. As reviewer pointed out, the image captions are hard to be read due to small sized letters. According to reviewer’s comment, we revised the manuscript by re-drawing the Figure 5B with more easily recognizable caption letters.

Comment 7:

6) Figure 7(c), photos presented are very small as well, difficult to recognize, what is presented.

Answer to comment 7:

Thank you for your comment. As reviewer pointed out, the small sized photos are hard to be recognized, leading to poor description of the slot die coating method. According to reviewer’s comment, in order to enhance the visibility and readability of Figure 7, we revised the manuscript by enlarging the size of photos as well as more clearly re-writing the caption letters in Figure 7.

Comment 8:

7) Page 15, line 491. Authors mentioned nanowires and whiskers as two different types of structures. What is the difference between them?

Answer to comment 8:

Thank you for your comment. As reviewer pointed out, the difference between nanowires and whiskers was not explicitly described in the text. The whisker in the text means nanowhisker which is a type of filamentary crystal (whisker) with cross sectional diameter ranging from 1 to 100 nm and length to diameter ratio >100. In addition, as whiskers are crystalline materials with distinct crystal anisotropy of properties and an almost dislocation-free structure, they make conventional plastic strain mechanisms almost inapplicable to them and brings their strength close to the theoretical ultimate strength value of a given material. As shown in Figure R1, the (nano)whiskers have been regarded as kind of nanowire because of similar structure, properties and morphology [R1].

Figure R1. An SEM image of Sn whiskers grown on a eutectic SnCu [R1].

According to reviewer’s comment, we revised the manuscript by more specifically describing the new terminology (whiskers) in the text.

[R1] Tu, K.-N.; Suh, J.-O.; Wu, A.T.-C.; Tamura, N.; Tung, C.-H. Mechanism and Prevention of Spontaneous Tin Whisker Growth. Mater. Trans. 2005, 46, 2300-2308.

Comment 9:

8) Figure 11, page 17. Photo captures are very small and difficult to understand

Answer to comment 9:

Thank you for your comment. As reviewer pointed out, the small sized photos and captions are hard to be recognized, leading to poor readability on the description of conductivity measurement. According to reviewer’s comment, we revised the manuscript by enlarging the size of photos as well as more clearly describing the captions (or enlarging caption letters) in the Figure 11.

Comment 10:

I would recommend authors to revise their graphical content in way to check that all images, and especially photos are easy to read. Also, there are some typos and minor English mistakes, that authors need to check before the final publication of the paper. Nevertheless, this paper presents a very detailed review of the field of Ag nanowire networks and should be published in Materials journal after minor corrections, listed above.

Answer to comment 10:

Thank you for your comment. As reviewer pointed out, we need to have many things revised by correcting/updating the graphical content (photos and captions in poor vision), typos and English mistakes. According to reviewer’s comment, we revised the manuscript by addressing the points as well as fully revise the full manuscript by professional English editing services. We believe the quality of our manuscript has been much improved after careful revision, being ready for published in “Materials

Round 2

Reviewer 1 Report

Dear authors,

After addressing all concerns of the reviewers, your manuscript is now ready for publication. I recommend it for publication as it is. Good luck and success for your future work.